# Frequency doubling in the cyanobacterial circadian clock

Bruno MC Martins[1,†], Arijit K Das[1,†], Liliana Antunes[1,2] & James CW Locke[1,3,4,*] iD

## Abstract

Organisms use circadian clocks to generate 24-h rhythms in gene expression. However, the clock can interact with other pathways to generate shorter period oscillations. It remains unclear how these different frequencies are generated. Here, we examine this problem by studying the coupling of the clock to the alternative sigma factor *sigC* in the cyanobacterium *Synechococcus elongatus*. Using single-cell microscopy, we find that *psbAI*, a key photosynthesis gene regulated by both *sigC* and the clock, is activated with two peaks of gene expression every circadian cycle under constant low light. This two-peak oscillation is dependent on *sigC*, without which *psbAI* rhythms revert to one oscillatory peak per day. We also observe two circadian peaks of elongation rate, which are dependent on *sigC*, suggesting a role for the frequency doubling in modulating growth. We propose that the two-peak rhythm in *psbAI* expression is generated by an incoherent feedforward loop between the clock, *sigC* and *psbAI*. Modelling and experiments suggest that this could be a general network motif to allow frequency doubling of outputs.

**Keywords** circadian clock; cyanobacteria; mathematical modelling; network motifs; single-cell time-lapse microscopy
**Subject Categories** Network Biology; Quantitative Biology & Dynamical Systems
**Mol Syst Biol. (2016) 12: 896**

## Introduction

Most organisms have evolved a circadian clock to anticipate the earth's cycles of light and dark (Doherty & Kay, 2010). The clock can drive downstream gene expression throughout the day and night with a 24-h rhythm. In mammals, the clock modulates multiple processes from sleep/wake activity to feeding cues (Mohawk *et al*, 2012). In plants, the clock can control photosynthesis, leaf movement and hypocotyl elongation (McClung, 2006; Millar, 2015).

Eukaryotic and prokaryotic algae have also evolved a circadian clock to coordinate downstream processes (Cohen & Golden, 2015; Noordally & Millar, 2015).

Circadian clocks can generate rhythms with periods other than 24 h. Recent work has revealed two peaks of activation per circadian cycle in a subset of genes in the mammalian suprachiasmatic nucleus (Pembroke *et al*, 2015). 12-h oscillations have also been observed in peripheral mammalian tissues, including the liver (Hughes *et al*, 2009; Cretenet *et al*, 2010), bone marrow (Chen *et al*, 2000) and testis (Liu *et al*, 2007) in mice. The circadian clock also appears to play a role in generating the approximately 12-h circa-tidal rhythm observed in some marine animals (Wilcockson & Zhang, 2008; Zhang *et al*, 2013). Computational analysis has suggested that joint regulation by two circadian factors with distinct phases could enable frequency doubling, with an over-representation of predicted binding sites for pairs of circadian factors found in the promoters of genes that display 12-h oscillations in mammals (Westermark & Herzel, 2013). Indeed, the product of pairs of circadian sinusoidal terms yields 12-h harmonics when the amplitudes of the two inputs are similar and their phases meet certain conditions (Korenčič *et al*, 2012; Westermark & Herzel, 2013). However, it remains unclear how gene circuitry couples with the circadian clock to generate 12-h rhythms *in vivo*.

The circadian clock circuit in the cyanobacterium *Synechococcus elongatus* PCC 7942 is an ideal model system to address the question of how the clock can generate such complex downstream gene expression. The core clock network is compact and well characterised (Johnson *et al*, 2011). Although simple, the clock coordinates key cellular processes throughout the day and night (Markson *et al*, 2013), including cell division (Dong *et al*, 2010), metabolism (Pattanayak *et al*, 2014; Diamond *et al*, 2015) and photosynthesis (Markson *et al*, 2013; Cohen & Golden, 2015). Nearly all transcripts in *Synechococcus elongatus* oscillate with a circadian rhythm (Liu *et al*, 1995; Ito *et al*, 2009). An additional advantage of this model system is that clock activity can be quantitatively analysed at the level of individual cells, using time-lapse microscopy of fluorescent reporters (Chabot *et al*, 2007; Dong *et al*, 2010; Teng *et al*, 2013).

The key components of the circadian clock in cyanobacteria have already been elucidated. The core network consists of just three

1   Sainsbury Laboratory, University of Cambridge, Cambridge, UK
2   Wellcome Trust Sanger Institute, Wellcome Trust Genome Campus, Hinxton, Cambridge, UK
3   Department of Biochemistry, University of Cambridge, Cambridge, UK
4   Microsoft Research, Cambridge, UK
    *Corresponding author. Tel: +44 1223 761110; E-mail: james.locke@slcu.cam.ac.uk
    †These authors contributed equally to this work

proteins (KaiA, KaiB and KaiC) that generate a 24-h oscillation in KaiC phosphorylation (Ito *et al*, 2007; Rust *et al*, 2007; Cohen & Golden, 2015). This oscillation has even been reconstituted *in vitro* (Nakajima *et al*, 2005). The phase of the clock is transmitted to downstream gene expression by the global regulator RpaA (Markson *et al*, 2013). Global analysis of target genes of the clock has revealed populations of genes that peak at dawn or dusk (Vijayan *et al*, 2009). However, it remains unclear what range of circadian gene expression profiles can be generated in cyanobacteria and by what mechanisms.

*Synechococcus elongatus* and other cyanobacteria have multiple group 2 sigma factors (Gruber & Gross, 2003), which have been suggested to provide a mechanism for modulating clock target gene expression. For example, the key photosynthesis gene *psbAI* has been shown to be regulated by both the clock and the alternative sigma factor *sigC* (Nair *et al*, 2002). *psbAI* encodes one form of the protein D1 (D1:1), which is essential for the initiation of photosynthesis and lies at the core of the photosystem II complex (Golden, 1995). A *sigC* deletion was shown to cause period lengthening of *psbAI* rhythms, but not of the KaiC oscillator, suggesting the possibility of multiple oscillators in one cell (Nair *et al*, 2002).

In this work, we examine a reporter for *psbAI* activity at the single-cell level through time. We reveal a doubling of the frequency of peaks in *psbAI* expression compared to the circadian clock, going from one to two peaks per circadian cycle. This two-peak oscillation was likely obscured in previous studies due to bulk averaging effects. We also observe two peaks of elongation rate per day, suggesting a physiological role in controlling growth for the frequency doubling. We go on to show through a combination of experiment and modelling that this two-peak oscillation is driven by an incoherent feedforward loop between the KaiC oscillator and *sigC*. We show that this network allows modulation of *psbAI* expression levels in low light conditions, and also find another gene, *rpoD6*, whose behaviour is modulated by the same network. Our work suggests that the cyanobacterial clock can couple with multiple downstream gene circuits to enable the generation of complex gene regulation.

# Results

### Time-lapse movies of *psbAI* expression reveal two-peak circadian oscillations

To analyse *psbAI* expression dynamics, we constructed a reporter strain incorporating a yellow fluorescent reporter (*yfp*) for *psbAI* expression and used quantitative time-lapse microscopy to examine $P_{psbAI}$-YFP levels in single cells. The *yfp* was tagged with a degradation tag (LVA) to enable tracking of fast cellular dynamics (Chabot *et al*, 2007). We first measured *psbAI* expression dynamics under constant 15 μE m$^{-2}$ s$^{-1}$ light conditions (Materials and Methods). Our time-lapse movies revealed that $P_{psbAI}$-YFP expression oscillates with a double-peak oscillation at the single-cell level (Figs 1A and EV1, and Movie EV1). Our single-cell approach was critical in revealing the double peak. Due to desynchronisation between cells, if we averaged all traces from three experiments (10 movies in total) the double peak was obscured (Fig 1A, red dashed trace).

We next examined the nature of the second peak in more detail. We carried out an analysis of peak-to-peak distances within each lineage. A distance of ca. 24 h represents a single peak per circadian cycle. However, we observe that a second peak of $P_{psbAI}$-YFP expression often occurs approximately 9 h after the first peak, although there is variability from cell to cell (Fig 1B). We also found that the second peak is, on average, smaller in amplitude than the first (Fig 1C).

### *sigC* is required for two-peak oscillations of *psbAI* expression

Previous work has shown that as well as being regulated by the circadian clock, *psbAI* is regulated by the alternative sigma factor *sigC* (Nair *et al*, 2002). To test whether *sigC* is responsible for the two-peak oscillation in *psbAI* expression that we observed, we generated a $P_{psbAI}$-YFP reporter in a *sigC* deletion background and repeated our time-lapse analysis. In this mutant background, we observe a single circadian oscillation in *psbAI* expression (Fig 1D). The single-cell traces display similar dynamics to the average trace from all experiments (Fig 1D, blue dashed, black and green lines), and peaks of expression occur on average 24 h apart in the majority of single cells (Fig 1E). The amplitude of the $P_{psbAI}$-YFP oscillations increased in the *sigC* deletion background, consistent with *sigC* negatively regulating *psbAI* expression (Fig 1F; Nair *et al*, 2002).

### Double peaks of *psbAI* expression are correlated with double peaks in cellular growth

We next asked whether we could observe any physiological effect of the double peak of *psbAI* expression. *psbAI* encodes the D1:1 form of the photosystem II reaction centre protein D1, which is the dominant form of the photosystem under the light levels used in this study (Clarke *et al*, 1993). As the D1:1 protein is unstable (Clarke *et al*, 1993), we hypothesised that two peaks of *psbAI* expression could result in two peaks of D1:1, and so correlate with two peaks in growth per circadian cycle. To test this, we calculated instantaneous single-cell elongation rates in both wild-type and *sigC* deletion backgrounds (Materials and Methods). *Synechococcus elongatus* cells are rod shaped, growing essentially only in one direction, and so the elongation rate is a good proxy for the cellular growth rate. Although variable, the mean elongation rate for each movie shows evidence of *sigC*-dependent double peaks in growth (Appendix Fig S1). To investigate this further, we computed the auto-correlation function of the single-cell elongation rates for each movie. The auto-correlation function peaks at lags of *ca.* ± 24 h, and also at lags of ± 12 h in the wild-type background (Fig 2A), providing evidence of two peaks of circadian growth per day. However, in the *sigC* deletion background, the auto-correlation function only peaks at lags of ± 24 h (Fig 2C), showing that there is a single peak of growth per circadian cycle.

To further understand the link between *psbAI* expression and growth, we computed the cross-correlation function (Dunlop *et al*, 2008) between the expression rate (obtained by differentiating the $P_{psbAI}$-YFP signal) and the elongation rate. We focus on the expression rate to be able to examine correlations on a fast timescale without the confounding factor of the degradation rate of the YFP reporter. The cross-correlation function peaks twice per circadian cycle and is near symmetrical in the wild type, showing the

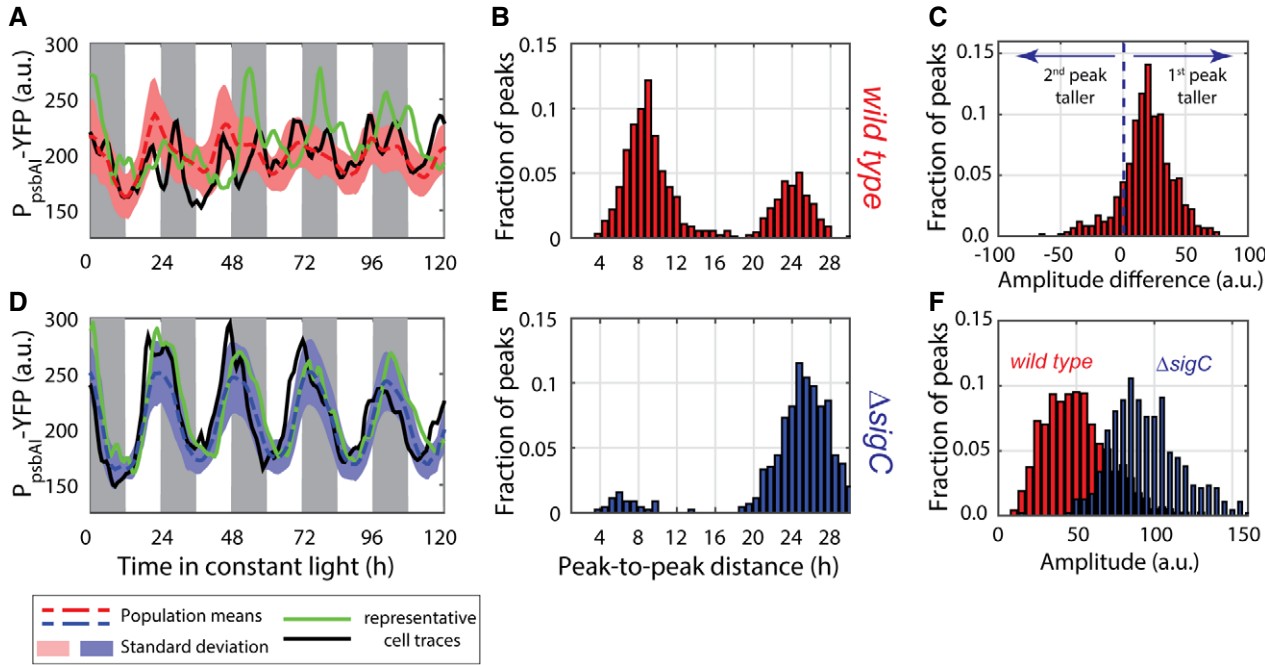

**Figure 1.  *psbAI* expression shows two peaks per circadian cycle.**

A  Time traces of P$_{psbAI}$-YFP reporter grown under constant low light (ca. 15 µE m$^{-2}$ s$^{-1}$ cool white light). Individual lineages (black and green lines) show the existence of a second peak following the first (dusk timed) peak of expression. Owing to cell-to-cell desynchronisation, this second peak is often hidden when expression is measured at the population level (dashed red line, with pink shades representing one standard deviation from the mean). The white and grey shades represent subjective day and subjective night, respectively. 1,319 cells from 10 movies (with up to 419 cells per time point) were collected.

B  Measure of the distance between the first peak in each circadian cycle and the following peak. The majority of peak pairs occur in the same circadian cycle, with a mean peak-to-peak distance of ca. 9 h within this subpopulation.

C  Distribution of the difference in peak amplitudes between the first and second peaks in a circadian cycle (for cycles where a double peak is present). The second peak is, on average, smaller in amplitude than the first peak.

D  Neither single-cell traces nor a population average of *psbAI* expression shows a double peak in a *sigC* deletion strain (lines and shades as in A). 1,088 cells from eight movies (with up to 419 cells per time point) were collected.

E  Measure of the distance between the first peak in each circadian cycle and the following peak shows that the vast majority of lineages have one peak of expression per day.

F  Distribution of peak amplitudes in wild-type and *sigC* deletion strains shows that *sigC* negatively regulates *psbAI* expression.

Source data are available online for this figure.

expression rate of *psbAI* and the elongation rate are positively correlated with only a very small lag (Fig 2B). The cross-correlation function peaks about 1 h before the origin (zero lag), implying significant changes in *psbAI* expression are reflected in changes in the growth rate 1 h later (Fig 2B, inset). We observe a similar pattern of temporal correlation between the two rates in a *sigC* deletion background, but only one peak per circadian cycle is present (Fig 2D). This suggests that two-peak oscillations in *psbAI* expression could play a functional role by tuning downstream processes that affect elongation rate.

### *sigC* shows a single circadian peak of expression at the single-cell level

Due to *sigC*'s role in generating the second peak in *psbAI* expression, we asked whether double peaks are already present at the level of *sigC* expression. We constructed a reporter for *sigC* expression (P$_{sigC}$-YFP) and repeated our analysis at the single-cell level. This reporter was also tagged with a degradation tag (LVA) (Chabot *et al*, 2007). We found that P$_{sigC}$-YFP expression oscillates with a

single circadian peak, and with single-cell traces that display similar dynamics to the average trace from all experiments (Fig 3A, red dashed, black and green lines, Appendix Fig S2 and Movie EV2). Peaks in *sigC* expression occur 24 h apart (Fig 3B). We also examined P$_{sigC}$-YFP dynamics in a *sigC* deletion background. Again, we observed a single circadian peak, the amplitude of which was approximately 3.5 times larger than in the wild-type case (Figs 3C and D, and EV2A). This fold change implies that SigC negatively regulates its own expression, reducing the amplitude of the P$_{sigC}$-YFP oscillation.

To test whether our observed dynamics of *psbAI* are solely due to *sigC* and the circadian clock, we next examined P$_{psbAI}$-YFP dynamics in a clock deletion background, and in a clock-*sigC* double deletion background. Clock deletion strains were generated by disrupting the *kaiBC* operon, which encodes the proteins KaiB and KaiC. No oscillation remains in mean P$_{psbAI}$-YFP levels in these lines (Appendix Fig S3A), suggesting that the dominant regulation of *psbAI* is through *sigC* and the clock. However, we do observe desynchronised fluctuations in *psbAI* expression in the *kaiBC* deletion backgrounds at the single-cell level (Appendix Fig S3B and C). The

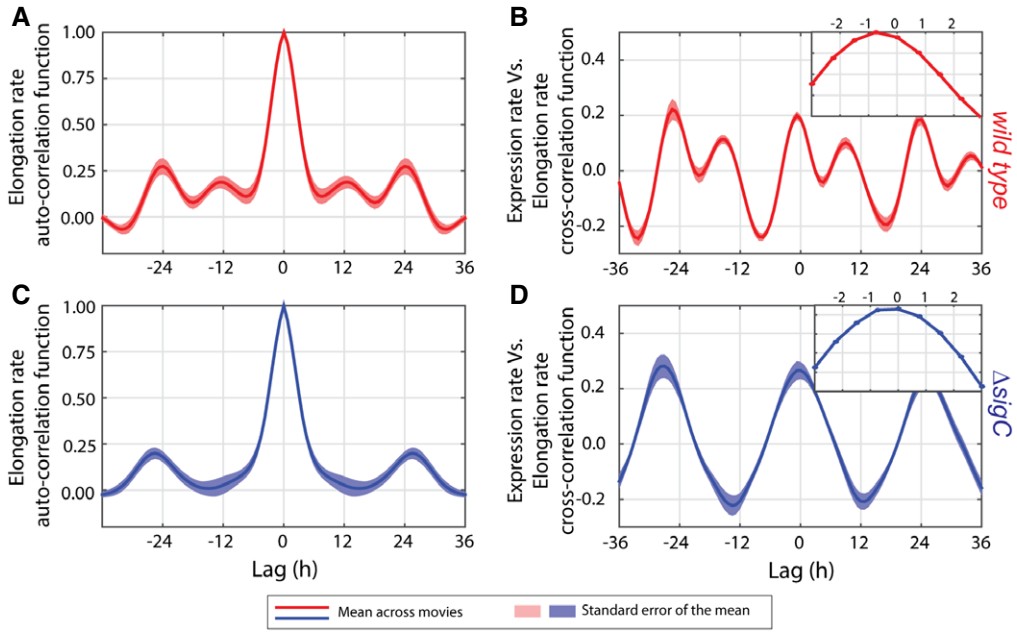

**Figure 2.  Two-peak circadian oscillations in growth can be observed in wild-type cells.**

A       Mean auto-correlation function of the elongation rate from nine movies (we removed one movie from the analysis as it ended after just 80 h) of the strain carrying the $P_{psbAI}$-YFP reporter in a wild-type background shows daily double peaks.

B       Mean cross-correlation function between expression rate of $P_{psbAI}$-YFP and elongation rate shows daily double peaks and a ca. 1-h delay of the elongation rate relative to the expression rate (inset with zoom-in of the mean trace between lags of −3 h and 3 h).

C, D    Both the mean auto-correlation function of the elongation rate (C) and the cross-correlation function between expression rate and elongation rate (D) from eight movies in a *sigC* deletion background show a single peak per circadian cycle. However, the elongation rate has a similar delay of ca. 1 h relative to the expression rate (D, inset).

Data information: Pink and light blue shades represent one standard error of the mean across all movies.

fluctuations of *psbAI* expression in clock deletion backgrounds are due to a dip of expression around the time of cell division (Appendix Fig S4). This suggests that the circadian clock can either damp or override cell cycle-related fluctuations in gene expression.

**An oscillatory feedforward loop generates two-peak oscillations in *psbAI* expression**

Based on our perturbation experiments, we hypothesised that an oscillatory feedforward loop between *sigC* and the circadian clock could be generating the observed dynamics (Fig 4A). The clock regulates both *psbAI* (Appendix Fig S3) and the intermediate factor SigC (Fig 3A), which also regulates *psbAI* (Fig 1D–F). The two arms of this joint regulation have opposite signs—the clock promotes *psbAI* expression whilst *sigC* inhibits it—thus forming an incoherent feedforward loop (Milo *et al*, 2002). Previous theoretical work has suggested that similar motifs, based on regulation by two factors, could enable frequency doubling of expression peaks in outputs of the circadian clock (Westermark & Herzel, 2013). We asked, with a simple mathematical model, whether a circuit consisting of an oscillatory feedforward loop between *sigC* and the clock and an autoregulatory feedback loop on *sigC* (Fig 3) could generate the observed dynamics of *psbAI* expression. To do this, we constructed a minimal phenomenological model of the system. The model consisted of a sinusoidal curve representing the output of the circadian clock,

$$\Theta(t) = b + \frac{1}{2}(A - b)(1 + \cos(\omega t)), \tag{1}$$

(where $b$ is the basal level of the clock signal, $A$ is its maximum, and $\omega$ is the angular frequency), and Hill equation terms for the activation dynamics of *sigC* and *psbAI* (Fig 4A). The ordinary differential equations for these two species take the general form

$$\frac{dx}{dt} = f_x - \Upsilon x, \tag{2}$$

where $\Upsilon$ is the dilution-degradation rate and $f_x$ is the production rate, which is defined as

$$f_x = V_x \frac{u_1^{h_1}}{1 + u_1^{h_1} + u_2^{h_2} + u_1^{h_1}.u_2^{h_2}}. \tag{3}$$

In equation (3), $V_x$ is the maximal production rate, $u_1$ and $u_2$ are regulatory inputs, and $h_1$ and $h_2$ are Hill coefficients. In the model, we assume that the negative regulation of *sigC* on *psbAI* and on itself is direct through an active form of SigC (see Section II in Appendix for full details of the model), although it is likely that there are intermediate steps given that sigma factors normally positively regulate transcription. The model was able to produce the qualitative behaviour seen in experiments, for both wild-type and mutant backgrounds (Fig 4B–D). We next tested what role the *sigC* auto-regulatory feedback loop plays in the dynamics. Unlike the regulation of SigC on *psbAI*, in our model the feedback of *sigC* on

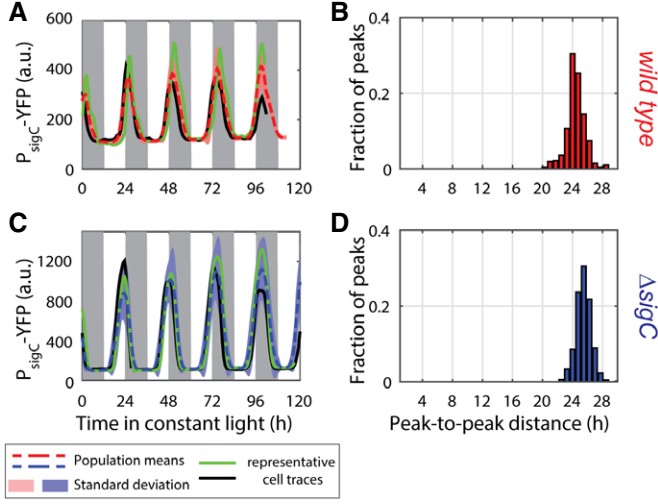

**Figure 3. *sigC* single-cell traces do not show a double peak, but show that *sigC* is negatively auto-regulated.**

A   Neither single-cell traces nor a population average of a reporter for *sigC*, $P_{sigC}$-YFP, shows a double peak of expression. 664 cells from nine movies (with up to 213 cells per time point) were collected.

B   Measure of the distance between the first peak in each circadian cycle and the following peak confirms that there is only one peak per circadian cycle.

C   The amplitude in *sigC* expression is raised by 3.5-fold in a *sigC* knock-out strain, consistent with a *sigC*-negative auto-regulation. 1,084 cells from six movies (with up to 352 cells per time point) were collected.

D   Measurement of the distance between the first peak in each circadian cycle and the next peak confirms that there is only one peak per circadian cycle.

Data information: Pink and light blue shades represent one standard deviation from the mean. Black and green lines represent two single-cell traces.
Source data are available online for this figure.

itself is not required to generate the two-peak oscillation (Section III in Appendix), but it adds an extra level of control to the circuit, allowing it to generate double peaks of expression more easily (Appendix Fig S5).

## Light levels modulate the strength of the second peak in *psbAI* expression

We next examined whether environmental conditions affect the double peak in $P_{psbAI}$-YFP, and whether this could be accounted for by the model. We examined the behaviour of our single-cell reporters for *psbAI* and *sigC* under higher constant light conditions (ca. 35 µE m$^{-2}$ s$^{-1}$), and again tracked single-cell behaviour. We found that under these higher light conditions the *psbAI* waveform was closer to a single-peak rather than a two-peak oscillation (Fig 5A). We carried out an analysis of peak-to-peak distances within each lineage and found the dominant mode was one peak per circadian cycle (Fig 5B). The *sigC* deletion had little effect on the dynamics of *psbAI* expression under these conditions (Fig 5C and D). This suggests that the regulation of *psbAI* by *sigC* is attenuated at higher light levels, resulting in single-peak oscillations.

We also observed that the expression of $P_{sigC}$-YFP is increased under these light levels (Fig EV2), consistent with a reduction in the repressive activity of *sigC*. If we model a reduction in SigC activity in our simple model, we also observe a reduction in the strength of

the second peak in *psbAI* (Fig EV3). This suggests that the balance between the two arms of the incoherent feedforward loop can set the strength of the two-peak oscillation. To confirm that the two-peak oscillation in *psbAI* is not specific to 15 µE m$^{-2}$ s$^{-1}$, we also carried out time-lapse microscopy of $P_{psbAI}$-YFP at lower light levels of *ca.* 10 µE m$^{-2}$ s$^{-1}$. Again, we observed a double peak in *psbAI* expression (Appendix Fig S6). Our work suggests that the multi-frequency oscillation in *psbAI* expression could allow sensitive regulation of photosynthesis at low light levels.

## An oscillatory feedforward loop can be a general motif to allow control of complex downstream gene expression

Our simulations suggest that the oscillatory feedforward motif should be able to generate a range of output dynamics, from 24-h single-peak oscillations to 12-h two-peak oscillations of near equal size (Appendix Fig S7). Between these two extremes, the simulations suggest that the strength of the double peak can be modulated by varying regulatory parameters, such as the thresholds and degrees of cooperativity in transcriptional activation. These parameters may therefore vary between different promoters. To test the generality of the motif, we examined the single-cell dynamics of the other group 2 sigma factors in cyanobacteria to test for signs of a two-peak oscillation. We chose the group 2 sigma factors as a starting point because sigma factors are known to cross-regulate (Nair *et al*, 2002; Imamura & Asayama, 2009). We observed double peaks in the expression dynamics of one of the group 2 sigma factors, *rpoD6* (using a $P_{rpoD6}$-YFP reporter line with the fast degradation LVA tag), at the single-cell level (Fig 6A). However, when compared to the double peak in *psbAI* expression (Fig 1A), the second peak in *rpoD6* expression was weaker (Fig EV4), often degenerating into a single-peak oscillation with a shoulderlike feature. These dynamics can be captured in our model by changing a parameter that affects only the interaction between the clock and *rpoD6* (Fig 6C). We repeated our analysis in a *sigC* deletion strain, and again observed that the multi-peak modulation is *sigC* dependent (Fig 6B and C). This suggests that the oscillatory feedforward loop motif can be a general mechanism for peak modulation and frequency doubling in circadian clocks.

# Discussion

Here, we report a novel dynamic behaviour of the cyanobacterial circadian clock, frequency doubling of outputs, and reveal a surprisingly simple and general mechanism for generating it. Using quantitative single-cell microscopy and a fluorescent reporter for *psbAI* expression, we found that *psbAI* is expressed with a two-peak circadian oscillation under constant low light conditions (Fig 1). This type of modulation of clock frequency has been observed in downstream outputs of the more complex mammalian circadian clock, although the mechanism remains elusive (Hughes *et al*, 2009; Westermark & Herzel, 2013; Pembroke *et al*, 2015). At the population level, these dynamics can be obscured due to desynchronisation, emphasising the value of examining gene circuit dynamics at the single-cell level (Martins & Locke, 2015; Fig 1A). We observe two circadian peaks of growth (Fig 2), as well as *psbAI* expression, showing a physiological effect of the frequency doubling in gene expression.

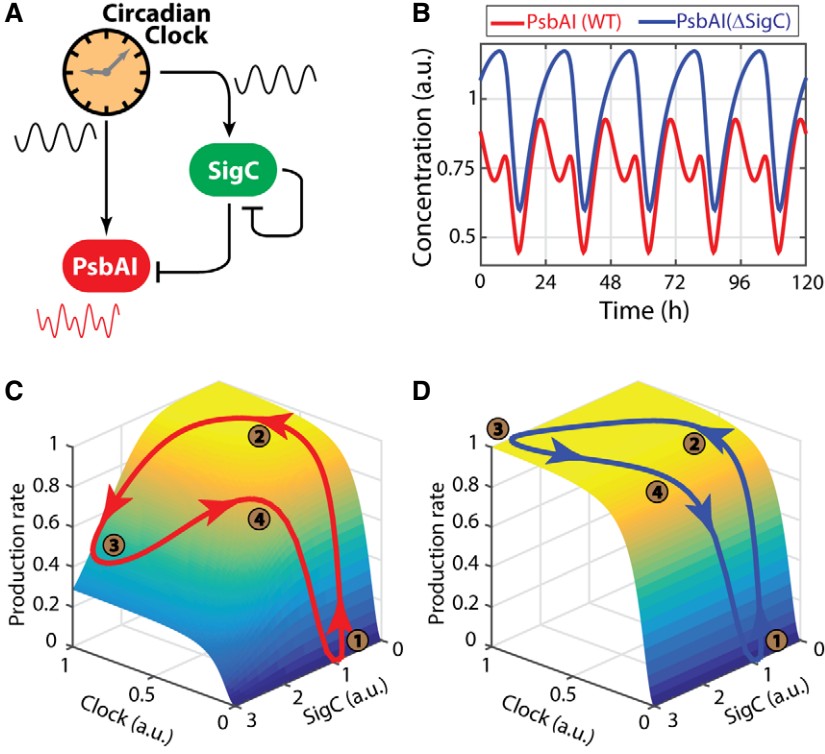

**Figure 4.  A minimal mathematical model, containing an incoherent feedforward loop modulated by an oscillatory signal, reproduces the experimental observations.**

A   Schematics of the proposed regulatory network. The circadian clock regulates both PsbAI and SigC, and SigC represses both PsbAI and itself. Since the circadian clock regulates PsbAI through two separate branches of opposite signs, one of which is mediated by an intermediate gene, this network contains an incoherent feedforward loop motif.

B   Numerical simulations of the wild-type network show double peaks of expression (red line), and numerical simulations of a SigC knock-out model (in which the terms representing the regulation of PsbAI by SigC are set to zero) show only single-peaked oscillations (blue line).

C   The trajectory of the normalised production rate of PsbAI in the wild-type (red line) simulation shows the dependence of the double peak on the states of the clock and SigC. The surface represents the two-input production rate function. When the levels of both the clock and SigC are low, the production rate is also low (time point 1, brown circle). As the state of the clock rises, the production rate follows and reaches a plateau (time point 2, which corresponds to the main peak of PsbAI in panel B). However, SigC also rises, crossing a threshold and overcoming the clock, thus imposing a trough in the production rate (time point 3). Later, SigC drops low enough to relieve its negative regulatory activity, and the production rate reaches a second peak (time point 4).

D   In the SigC deletion simulation (the terms representing the regulation of PsbAI by SigC are set to zero, but SigC expression is tracked for reference), the production rate of PsbAI only responds to the clock, which is a single-peak oscillation, and so the trajectory of the production rate (blue line) only has a single peak itself (the plateau where time points 2, 3 and 4 lie).

The complex gene regulation of *psbAI* can be generated by a simple gene circuit motif (Fig 4). An oscillatory incoherent feedforward loop between the circadian clock, *sigC* and *psbAI* allows the 24-h oscillator to generate a two-peak circadian oscillation in its output genes. The incoherent feedforward loop is one of the most common network motifs, appearing hundreds of times in bacteria and yeast (Milo *et al*, 2002; Shen-Orr *et al*, 2002). Under a steplike input, the incoherent feedforward motif has been shown to generate a pulse in downstream gene expression (Mangan & Alon, 2003; Basu *et al*, 2004). It can also enable fold change detection in gene regulation (Goentoro *et al*, 2009). Although these functions of feedforward loops have been analysed, the role of feedforward loops as modulators of oscillations is less explored, and only suggested theoretically (Cerone & Neufeld, 2012). In this work, we identified experimentally a new function for the incoherent feedforward loop in enabling frequency doubling. It will be exciting to see the range of uses cells can make for this and related motifs in an oscillatory context.

The two-peak oscillation in *psbAI* expression is modulated by light conditions (Fig 5), becoming more apparent under low light. Previous single-cell bioluminescence studies using a luciferase reporter driven by the *psbAI* promoter displayed single-peak oscillations under higher light conditions (100 μE m$^{-2}$ s$^{-1}$) (Mihalcescu *et al*, 2004), similar to what we observe under 35 μE m$^{-2}$ s$^{-1}$ light. However, in bioluminescence assays the signal for each data point is integrated over long windows during which ambient illumination is turned off (leading to lights being off for a large fraction of the day), so the conditions are not directly comparable to ours. The dynamic behaviours of the *sigC*-clock motif appear therefore to be tuned not just by regulatory interactions, but also by the external environment. Further work should be carried out to understand how the motif is modulated by light. One hypothesis is that a reduction in SigC activity at higher light results in a reduction in the double peak. This idea is supported by experiments and our model (Figs EV2 and EV3), but requires

                    

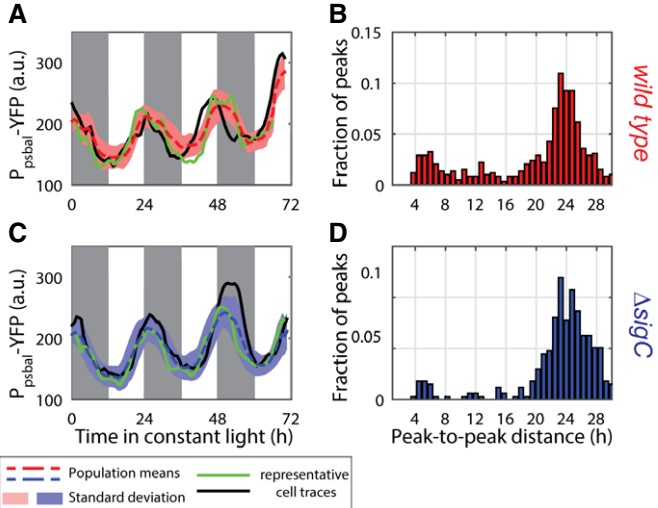

**Figure 5. The double peak in expression of *psbAI* is largely lost at higher light conditions.**

A Time traces of P$_{psbAI}$-YFP reporter grown under ca. 35 µE m$^{-2}$ s$^{-1}$ cool white light. Most individual lineages (black and green lines) show one circadian peak of gene expression. 2,601 cells from eight movies (with up to 776 cells per time point) were collected.

B Measure of the distance between the first peak in each circadian cycle and the following peak shows a single peak per day is the dominant mode.

C Time traces of P$_{psbAI}$-YFP reporter in a *sigC* deletion background look similar to the wild type. 2,306 cells from five movies (with up to 623 cells per time point) were collected.

D Measure of peak-to-peak distance shows a single circadian peak.

Data information: Pink and light blue shades represent one standard deviation from the mean.

Source data are available online for this figure.

further validation. Interestingly, light levels much higher than those used in our study resulted in the degradation of *psbAI* transcripts (Kulkarni *et al*, 1992). It will be important to observe how the clock-*sigC* circuit interacts with other genetic and environmental modifiers of *psbAI* (Nair *et al*, 2002; Thomas *et al*, 2004).

Our observation that wild-type cells can show two-peak circadian oscillations in growth (Fig 2A and B) suggests a functional significance of frequency doubling, as the clock-*sigC* circuit could enable tuning of growth under different environmental conditions. The circadian clock has been shown to have an adaptive benefit under light–dark cycles (Woelfle *et al*, 2004). Although our work focuses on the general mechanism of how frequency doubling of expression peaks can be generated, in future it will also be important to examine any possible adaptive benefit of the two-peak oscillation in *psbAI* expression, under both varying levels of constant light and light–dark cycles.

*psbAI* encodes one form of the photosystem II reaction centre protein D1 (D1:1). Currently, it is difficult to probe D1:1 protein dynamics in single cells due to the perturbative effects of fusing reporter proteins to proteins of interest and the lack of single-cell proteomics techniques in cyanobacteria. However, our observation of two peaks of circadian growth in the wild-type background suggests that double peaks in D1:1 levels, as well as *psbAI* transcripts, could be possible in individual cells, particularly in the light of reports that the D1:1 protein is unstable, with a measured half-life

of approximately 50–60 min at 50 µE m$^{-2}$ s$^{-1}$ light (Clarke *et al*, 1993).

A simple mathematical model can recapture the dynamics observed (Fig 4). It suggests that modulation of some parameters, such as the production rate of the target gene, can modify the output oscillations of the feedforward loop from a two peak to a shoulder of gene expression (Fig 6D). Support for this model is provided by the fact that we can observe these two types of dynamics experimentally: two peaks in *psbAI* (Fig 1) and shoulder like in *rpoD6* (Figs 6 and EV4), both dependent on *sigC* (Figs 1B and 6C). A synthetic approach where we tune, for example, the production rate of a target gene by modifying its promoter sequence, and verify whether we can switch between the two dynamic behaviours, would be a further test of the model.

An expansion of the model may, however, be required in order to understand published reports of period lengthening in *sigC* knock-out strains (Nair *et al*, 2002). We also observed a small period lengthening effect in the *sigC* knock-out strain. In the wild-type background, we measured average peak-to-peak distances of 24.07 h for the single frequency mode of P$_{psbAI}$-YFP (Fig 1B) and 24.43 h for P$_{sigC}$-YFP (Fig 3B). In the *sigC* knock-out background, these average distances increase to, respectively, 25.66 h (Fig 1E) and 25.47 h (Fig 3D). This suggests there may be a feedback of SigC onto the clock. However, bulk measurements of expression of the *KaiBC* operon did not reveal period lengthening (Clerico *et al*, 2009) in *sigC* knock-out backgrounds. Additionally, both *KaiBC* and *sigC* were recently found to be a direct target of RpaA, a global circadian regulator that is one of the major outputs of the clock (Markson *et al*, 2013). If feedback from *sigC* onto the clock exists, then it must operate downstream of RpaA, and extra components in the clock-*sigC*-*psbAI* circuit wait to be revealed. These extra components will not affect the mechanism of frequency doubling we describe here, as the incoherent feedforward loop will still be embedded in the circuit and drive two-peak oscillations.

The cyanobacterial clock is seemingly much simpler than that of higher eukaryotes, which consists of multiple transcriptional and translational feedback loops (Bell-Pedersen *et al*, 2005). Our work suggests that although the core of the cyanobacterial clock consists of just three interacting proteins, the clock couples with other pathways to generate complex downstream gene expression. SigC is one of at least nine alternative sigma factors in *Synechococcus elongatus* (Imamura & Asayama, 2009). It will be interesting to observe whether the other alternative sigma factors couple with the clock to produce specific dynamics in downstream targets. It will also be important to test whether higher eukaryotic clocks use similar network motifs to generate frequency doubling of peaks of expression in downstream targets.

Increasingly, oscillations in gene regulation are being observed in diverse cellular processes, with a wide range of regularity and periods. These include pulses in stress response genes, from *sigB* in *B. subtilis* (Locke *et al*, 2011) to *p53* in mammals (Batchelor *et al*, 2011), and deterministic oscillations in genes, such as ERK, with nucleus to cytoplasm oscillations of 15 min (Shankaran *et al*, 2009). Understanding the rules of oscillatory gene regulation will become increasingly important. It will be interesting to observe whether other oscillatory circuits use similar circuit designs to modulate output frequency.

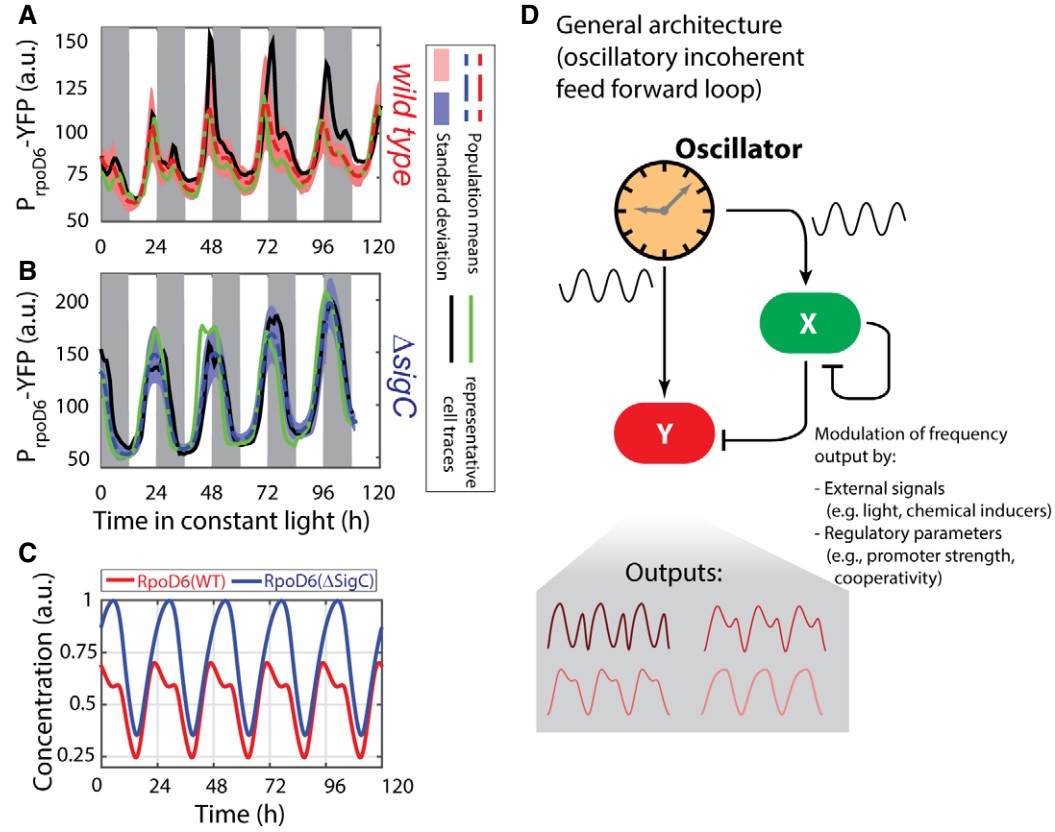

**Figure 6. The clock-*sigC* circuit represents a general mechanism to generate multi-peak oscillations in oscillatory networks.**

A Time traces of P$_{rpoD6}$-YFP reporter grown under low light conditions (ca. 15 µE m$^{-2}$ s$^{-1}$ cool white light). Individual lineages (black and green lines) show the existence of a shoulder or a secondary peak of reduced prominence when compared to P$_{psbAI}$-YFP. 947 cells from 12 movies (with up to 272 cells per time point) were collected. Pink shades represent one standard deviation from the mean.

B Time traces of P$_{rpoD6}$-YFP reporter in a *sigC* deletion background show only a smooth single-peaked oscillation. 1,352 cells from seven movies (with up to 502 cells per time point) were collected. Light blue shades represent one standard deviation from the mean.

C Numerical simulations of the RpoD6 wild-type network show a shoulder of expression trailing the main peak (red line). All the parameters describing the clock and SigC are as in Fig 4B, and only the threshold of activation of the *rpoD6* promoter by the clock was modified. Numerical simulations of a SigC knock-out model (in which the terms representing the regulation of RpoD6 by SigC are set to zero) show only single-peaked oscillations (blue line).

D The incoherent feedforward loop circuit that regulates *rpoD6* and *psbAI* is capable of generating diverse oscillatory dynamics *in vivo* and *in silico*. Networks where a target gene (Y) is co-regulated by an oscillator and another regulator (X, which itself is regulated by the oscillator) may represent a general mechanism for generating multi-peak oscillations.

# Materials and Methods

### Bacterial strains, plasmids and DNA manipulations

*Synechococcus elongatus* strains were derived from the ATCC strain (ATCC$^{®}$ 33912$^{™}$). Strains and plasmids used in this study are described in Tables 1 and 2, respectively. Briefly, a 500-bp upstream region of *psbAI*'s coding sequence, and 1,000-bp upstream regions of both *sigC*'s and *rpoD6*'s coding sequences were amplified using primers that incorporated restriction enzyme sites compatible for cloning into a NS1 targeting vector, pAM2314 (a kind gift from Prof. Susan Golden). The promoter regions were put upstream of an SsrA(LVA)-tagged YFP protein (Andersen *et al*, 1998; Chabot *et al*, 2007), which was constructed by adding a consensus LVA tag to the YFP coding region from plasmid pAM3685 (a kind gift from Prof. Susan Golden). Clock deleted versions of these reporter strains were generated by insertion of a

gentamicin resistance cassette into the ORF of the *kaiBC* operon (the plasmid was a kind gift from Prof. Erin O'Shea). The plasmid carrying the interrupted gene with the antibiotic marker was transformed into the reporter strains. Complete allele replacement on all the chromosomal copies was checked through PCR. Deletion of *sigC* was achieved in a similar manner using a kanamycin resistance cassette from the vector pUC19. Reporter strains carrying both inactivated *sigC* and *kaiBC* were generated by sequential disruption of these genes through homologous recombination, and verified through PCR. The antibiotic levels used for the culturing of strains were as follows: 2 µg ml$^{-1}$ spectinomycin/streptomycin for NS1-based pAM2314 inserts, 5 µg ml$^{-1}$ kanamycin for the *sigC* deletion mutants and 2 µg ml$^{-1}$ gentamicin for *kaiBC* deletion mutants.

In the time-lapse experiments, under conditions of low light (*ca.* 15 µE m$^{-2}$ s$^{-1}$), we measured a mean cell cycle time of 19.5 ± 4.6 h, which is similar to what has previously been

measured in other single-cell studies (Dong *et al*, 2010), and a mean cell elongation rate of $0.031 \pm 0.004$ h$^{-1}$. Cells grown under higher light (*ca.* 35 µE m$^{-2}$ s$^{-1}$) had a mean cell cycle time of $9.3 \pm 2.7$ h and a mean cell elongation rates of $0.07 \pm 0.01$ h$^{-1}$. Finally, cells grown under very low light (*ca.* 10 µE m$^{-2}$ s$^{-1}$) had a mean cell cycle time of $25.2 \pm 5.8$ h and a mean cell elongation rate of $0.025 \pm 0.004$ h$^{-1}$.

## Growth conditions

The strains were grown in BG-11 M media at 30°C under photoautotrophic conditions with constant rotation, supplemented with appropriate antibiotics. Light conditions were maintained at approximately 25 µE m$^{-2}$ s$^{-1}$ by cool fluorescent light sources. Before the start of each movie acquisition, the cultures were entrained by subjecting the cells to two 12-h light:12-h dark cycles (12:12LD).

## Microscopy and sample preparation

A Nikon Ti-E inverted microscope equipped with the Nikon Perfect Focus System module was used to acquire images of *Synechococcus elongatus* reporter strains. 2 µl of entrained cultures in exponential phase was diluted to an OD$_{750}$ of 0.15 and applied to agarose pads. The protocol followed was modified from Young *et al* (2012). The agarose pads were dried and placed inside a two-chambered cover glass (Labtek Services, UK). The microscopy set-up includes a custom-built circular cool white light LED array (Cairn Research, UK), attached around the condenser lens. Data acquisition was controlled through the software Metamorph (Molecular Devices, California). The samples were set in the microscope and further entrained for an additional LD cycle before starting data acquisition. The cells were kept under constant light conditions throughout the acquisition, except whilst taking fluorescent images. Epi-illumination was provided by a solid-state white light source (Lumencor, Oregon). Filter sets 41027 (Calcium Crimson) and 49003 (ET-YFP) by Chroma Technology, Vermont, were used. Following best practice in the field, we minimised fluorescent exposure time and intensity, and as frequencies of imaging faster than once every half hour are generally discouraged (Yokoo *et al*, 2015), phase contrast and fluorescent images were acquired every 45 min using a CoolSNAP HQ2 camera (Photometrics, Arizona). All experiments used a 100× objective.

## Quantification of fluorescence

The images were processed and analysed using custom-made software in MATLAB (The MathWorks, Massachusetts) adapted from Young *et al* (2012), for segmentation and single-cell tracking. Mean fluorescence per cell is defined as the average of the intensities of all pixels within the segmented area of each cell. For each movie, all single-cell traces were smoothed by a moving average filter spanning five data points (3 h). Whenever the range of the filter window is not contained within the range of a cell cycle (i.e. when the filter is centred in one of the first two or in one of the last two data points of each individual cell cycle), single-cell traces are padded with data from the mother (on the left) and the average of the two daughters (on the right), provided data for these cells have been acquired.

## Quantitative analysis

*Peak-to-peak measurements*
For each micro-colony, we extracted all single-cell lineages and detrended the data by subtracting a second-order polynomial fit. We then fitted a sinusoid to each lineage using a cosinor method (Nelson *et al*, 1979) constrained to the range [21 h, 27 h]. This fit finds the best *single*-period sinusoid for the time series, and we use it to define the characteristic circadian windows of each lineage. For lineage $i$, we therefore obtain a sinusoidal fit $F_i$ such that

$$F_i = \alpha + \beta \cos\left(\frac{2\pi(t + \phi_i)}{\tau_i}\right), \text{ with } 21 < \tau_i < 27. \qquad (4)$$

We then compute the troughs (minima) of $F_i$ and the time points $t_{i,j}$ when they occur. Next, we extract the mean fluorescence signal $Y_{i,[j,j+1]}$ between each pair $t_{i,j}$ and $t_{i,j+1}$. We loop through all $Y_{i,[j,j+1]}$ sub-series and obtain candidate peaks of expression using the standard function "findpeaks" in MATLAB. We impose all peaks must be at least 3 h apart and have a width of at least 2 h at full width half maximum. Additionally, if there is at least one peak remaining, we discard all other peaks with heights lower than 25% and prominences lower than 10% of the major peak, as well as those pairs with width differences greater than 5 h. This filtering is intended to eliminate spurious fluctuations and maintain only clear independent peaks. Peak-to-peak distances are then defined as the distance between two consecutive peaks within each $Y_{i,[j,j+1]}$

**Table 1. Strains used in this study.**

| *Synechococcus elongatus* strain | Genetic background | Reporter (Integration site) | Antibiotic resistance | Source |
|---|---|---|---|---|
| 7942_A1 | WT | None | None | ATCC |
| 7942_S19 | WT | P$_{psbAI}$::YFP_LVA (NS1) | Sp$^r$,St$^r$ | This study |
| 7942_A50 | ΔsigC | P$_{psbAI}$::YFP_LVA (NS1) | Sp$^r$,St$^r$,Kan$^r$ | This study |
| 7942_A52 | ΔkaiBC | P$_{psbAI}$::YFP_LVA (NS1) | Sp$^r$,St$^r$,Gent$^r$ | This study |
| 7942_A51 | ΔsigC,ΔkaiBC | P$_{psbAI}$::YFP_LVA (NS1) | Sp$^r$,St$^r$,Kan$^r$,Gent$^r$ | This study |
| 7942_A4 | WT | P$_{sigC}$::YFP_LVA (NS1) | Sp$^r$,St$^r$ | This study |
| 7942_A57 | ΔsigC | P$_{sigC}$::YFP_LVA (NS1) | Sp$^r$,St$^r$,Kan$^r$ | This study |
| 7942_A10 | WT | P$_{rpoD6}$::YFP_LVA (NS1) | Sp$^r$,St$^r$ | This study |
| 7942_A60 | *ΔsigC* | P$_{rpoD6}$::YFP_LVA (NS1) | Sp$^r$,St$^r$,Kan$^r$ | This study |

**Table 2.   Plasmids used in this study.**

| Plasmid | Description | Antibiotic resistance | References |
|---|---|---|---|
| pAM2314 | Cyanobacterial cloning vector with NS1 integration sequence | Sp$^r$, St$^r$ | Ditty *et al* (2005) |
| pAM3685 | P$_{trc}$::*kaiA*-YFP | Sp$^r$,St$^r$ | Dong *et al* (2010) |
| pLA35 | P$_{psbAi}$::YFP_LVA in pAM2314 | Sp$^r$,St$^r$ | This study |
| pAD03 | P$_{sigC}$::YFP_LVA in pAM2314 | Sp$^r$,St$^r$ | This study |
| pAD09 | P$_{rpoD6}$::YFP_LVA in pAM2314 | Sp$^r$,St$^r$ | This study |
| pUC18-*kaiBC*-Gent | *kaiBC* deleted by the insertion of gentamicin cassette within the ORF | Gent$^r$ | Teng *et al* (2013) |
| pAD35 | *sigC* inactivated by insertion of kanamycin cassette within the ORF. | Kan$^r$ | This study |

sub-series. However, if there is only one peak in the sub-series, then we measure the distance to the first peak in the next sub-series instead. We do not consider any peak-to-peak distances greater than 36 h. Finally, and because many sub-series are repeated in different lineages, we loop through all peak-to-peak pairs and discard all non-unique pairs.

*Elongation rate*

Due to being rod-shaped, cell length in *S. elongatus* is a good proxy for cell volume, and so elongation rates are correlated with single-cell growth rates. For each micro-colony, we extracted single-cell lengths across all frames. For each cell, we smooth the length traces and use a moving window to fit cell lengths to an exponential function. The width of the window is one-third of the average cell cycle duration in the micro-colony. When fitting for windows centred in early and late points in the cell cycle, we stitch the progenitor and offspring length traces, but normalising lengths to maintain continuity and obtain a monotonic length profile. In this way, we can average *across* division events and obtain a continuous function for the elongation rate, which is not interrupted at the end of each cell cycle. The instantaneous elongation rate at each time point is the exponent in the fitted function. Next, we averaged the single-cell traces across all cells for each micro-colony.

*Auto-correlations and cross-correlations*

We used the method developed by Dunlop *et al* (2008), which quantifies cross-correlation functions for tree-structured data by subtracting the contributions of data points that are common to more than one single-cell lineage. Briefly, we first extract all single-cell lineages in each movie. These lineages trace the time evolution of the variables of interest (YFP expression, elongation rate) across all cells within a lineage, from the beginning of the movie until its end. The normalised cross-correlation function between two variables, $x$ and $y$, is given by

$$R_{x.y}(\tau) = \frac{S_{x.y}(\tau)}{\sqrt{S_{x.x}(0)S_{y.y}(0)}}, \tag{5}$$

where $\tau$ is the lag between the two variables. The standard definition of $S_{x.y}(\tau)$ is

$$S_{x.y}(\tau) = \begin{cases} \frac{1}{N_t-\tau}\sum\limits_{n=0}^{N_t-\tau-1}\tilde{x}(n+\tau)\tilde{y}(n); & \tau \geq 0 \\ S_{x.y}(-\tau); & \tau < 0 \end{cases}, \tag{6}$$

where $\tilde{x}$ and $\tilde{y}$ are the two variables rescaled to have zero mean.

Next, the cross-correlations of all individual lineages within a movie are averaged. However, because two sister cells branch out from the same data point and, prior to this branching, they share the same time series, the cross-correlation functions are modified to discard all repeated data points. This ensures each pair of points for the two variables we are cross-correlating is only weighed once. The modified cross-correlations are given by

$$S_{x.y}(\tau) = \begin{cases} \frac{1}{N_t-|\tau|}\frac{1}{N_c}\left[ \begin{array}{l} \sum\limits_{i=0}^{N_c-1}\left(\sum\limits_{n=0}^{N_t-\tau-1}\tilde{x}_i(n+\tau)\tilde{y}_i(n)\right) \\ -\sum\limits_{i=0}^{N_c-2}\left(\sum\limits_{n=0}^{d_i-\tau-1}\tilde{x}_i(n+\tau)\tilde{y}_i(n)\right) \end{array} \right]; & \tau \geq 0 \\ S_{x.y}(-\tau); & \tau < 0 \end{cases} \tag{7}$$

Here $\tilde{x}_i = x_i - \frac{1}{N_c}\sum_{i=0}^{N_c-1}x_i$, $N_c$ is the number of cells in the movie, $N_t$ is the number of data points, and $d_i$ are the branching points (i.e. the times of cell division events) (Dunlop *et al*, 2008).

Finally, the cross-correlation functions of all movies are averaged, with all movies carrying the same weight.

**Mathematical model**

A minimal phenomenological ODE model of the incoherent feedforward loop circuit was constructed using a sinusoidal curve as a representation of the circadian clock, and Hill equation kinetics for *sigC* and *psbAI*. Equations were solved numerically using the stiff ode23s solver in MATLAB. The model is provided in SBML code (Code EV1). See Appendix Table S1 for model parameters, and section II in Appendix for further details of the modelling approach and assumptions.

**Data availability**

The model is provided in SBML code as Code EV1 and is also available at the JWS Online model database (https://jjj.bio.vu.nl/models/martins2/).

**Expanded View** for this article is available online.

**Acknowledgements**

We wish to thank Niall Murphy for help with converting the mathematical model into the sbml format and Chao Ye for help with setting experiments. We thank Susan Golden and Erin O'Shea for strains. We thank Pau Formosa-Jordan, Om Patange, Katie Abley, Michael Elowitz and Henrik Jönsson for critical reading of the manuscript and useful suggestions. This research was made possible by the award of a European Research Council under the European

Union's Seventh Framework Programme (FP/2007-2013)/ERC Grant Agreement 338060. The work in the Locke laboratory is further supported by a fellowship from the Gatsby Foundation (GAT3272/GLC) and a Fellowship from the Human Frontier Science Program (CDA00068/2012).

## Author contributions

BMCM, AKD and JCWL conceived and designed the study, analysed and interpreted the data, and wrote the article. BMCM and AKD performed the experiments. AKD and LA constructed strains. BMCM developed the mathematical model.

## Conflict of interest

The authors declare that they have no conflict of interest.

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
