## [Review Process File · Molecular Systems Biology]

Frequency doubling in the cyanobacterial circadian clock

Bruno Martins, Arijit Das, Liliana Antunes and James Locke

Corresponding author: James Locke, Cambridge University

Review timeline:

Submission date:	13 June 2016
Editorial Decision:	06 July 2016
Revision received:	27 October 2016
Editorial Decision:	16 November 2016
Revision received:	17 November 2016
Accepted:	23 November 2016

Editor: Maria Polychronidou

Transaction Report:

1st Editorial Decision

06 July 2016

Thank you again for submitting your work to Molecular Systems Biology. We have now heard back from the three referees who agreed to evaluate your study. As you will see below, the reviewers acknowledge that the presented findings are interesting. However, they raise a number of concerns, which should be carefully addressed in a revision of the manuscript. The recommendations of the referees are clear, so there is no need to repeat the points listed below.

REFeree REPORTS

Reviewer #1:

In this manuscript Martins, Das and Locke use YFP reporters and microscopy and to examine the nature of psbAI (paralogs should be roman) expression in single *Synechococcus elongatus* cells. They find that psbAI expression exhibits two peaks of gene expression at very low light that is dependent on both sigC and the circadian clock (both of which have been previously demonstrated to be required for proper expression of psbAI at the population level). These results were dependent on single cell analysis as averaging effects appear to mask this effect. Furthermore they demonstrate that rpoD6, a group 2 sigma factor, also displays a similar 2 peak expression pattern over a 24-h cycle, suggesting that 2 peaks of expression may be a more generalized mechanism. The authors also demonstrate that *S. elongatus* exhibits 2 peaks in cell elongation and try to make a correlation with psbA1 expression. Combined with mathematical modeling the authors put forward a model suggesting that the two peak oscillation are driven by an incoherent feedforward loop where the circadian clock activates both sigC and psbA1 and sigC in turn inhibits psbAI generating two peak oscillations in every 24-h cycle. While these data are interesting and shed light into how the circadian outputs can be modulated to generate rhythms of different periodicities there are some concerns regarding some of the data interpretation.

1. In particular there were several points of inconsistency regard the sigC results:
 - a. It was unclear how the traces in Figure 3 were generated for the P_{sigC}-YFP reporter. Movie 2, which showed these strains growing displayed a great deal of cell to cell variation. It was unclear as to how such variation observed in the movie resulted in the coherent and large amplitude rhythms shown in Fig. 3.
 - b. The authors show that psbAI rhythms in a sigC mutant background have a single peak of expression with a 24h period oscillation. However published results (Nair et al., 2002) show period lengthening. Can the authors explain this discrepancy?
 - c. There also appeared to be some confusion about the role for SigC in their model. In particular the authors mention in lines 178-181 that the SigC auto-regulatory feedback was not necessary to observe the two peak oscillation. However, their data (Figure 1D) suggests that sigC is absolutely necessary for this observation. How are these reconciled?
2. Additionally, the authors claim that psbAI levels can account for the 2 peaks in cell elongation. However, *S. elongatus* displays what is known as circadian gating of cell division, where the circadian clock specifically inhibits cell division for several hours around subjective dusk. Could the observed 2-peaks in cell elongation be due to circadian gating of cell division? Is this two peak in elongation rate observed in a kai mutant background?
3. The authors do not mention the fact that the O'Shea lab found the sigC gene to be a direct target of RpaA, which is the transcriptional master regulator by which the Kai oscillator controls gene expression. With this point in mind, can the authors be more specific about how the two loops interact?
4. Although not essential for this paper, it would have been useful to also test expression of a gene not known to be strongly affected by sigC or by environmental conditions, such as the kaiBC promoter. Data may be available that could be incorporated in the model. See:

Stability and lability of circadian period of gene expression in the cyanobacterium *Synechococcus elongatus*. Clerico EM, Cassone VM, Golden SS. *Microbiology*. 2009 Feb;155(Pt 2):635-41. doi: 10.1099/mic.0.022343-0. PMID: 19202112

Reviewer #2:

This is an interesting paper that seeks to explain how the circadian clock in the cyanobacterium *Synechococcus elongatus*, which beats on a 24 h basis, is also able to generate shorter period (12 h) rhythms by frequency doubling.

The authors created a transgenic strain that expresses YFP under the control of the psbA1 promoter, a key gene involved in photosynthesis, which has been used as a circadian marker in many guises before, mostly using bioluminescence. However, previous analyses have employed population assessment of bacterial psbA1 expression, rather than single cell analyses, and therefore have not revealed 12 h oscillations previously. The paper presents compelling data, by genetic deletion, that an important component mediating frequency doubling of psbA1 is the sigC sigma factor.

Conceptually, the paper presents a new model of how frequency doubling can occur in this particular case (and is modeled well by an incoherent feedforward loop), but also potentially in other genetic circuits (e.g. mammalian clock). Therefore, the study will be of interest to researchers interested in oscillatory phenomena, including those studying circadian clocks in many organisms.

Major points

1. The authors should discuss, and justify, why previous bioluminescence studies performed using single cell imaging (e.g. Mihalcescu et al. *Nature* 2004) did not observe 12 hour rhythms, even though psbA1 drove expression of bacterial luciferase in these cells.
2. Related to (1), the authors should exclude a significant effect of pH in determining the phenomenon they observe. YFP fluorescence is known to be significantly affected by pH. If, for example, sigC drives a 24 h rhythm in pH (that was independent of the psbA1 rhythm), this could produce a second peak in fluorescent that was not due to psbA1 at all. One way to resolve this might be to express YFP under a constitutive promoter and see whether YFP fluorescence oscillates. An alternative would be to assay YFP protein / transcript levels (or psbA1 itself) in synchronized psbA1-YFP cells. It would be expected that if 12 h and 24 h rhythms are driven by the psbA1

promoter, then this should be seen in expression patterns, in addition to YFP fluorescence.

Minor points

1. Does fluorescence excitation itself affect the clock in the cyanobacterial cells? This is important since light levels influence the phenomena shown in the manuscript. Citing previous literature would be sufficient, if available.
2. "Dampen" means "to wet". This should be changed to "damp".

Reviewer #3:

The manuscript combines in a convincing way large scale single cell imaging, appropriate reporters and mutations, network motifs and mathematical modeling. The key observations are frequency doublings (or harmonics) related to the circadian clock. In the introduction other interesting biological examples are discussed with 12 hour rhythms including circa-tidal rhythms. Thus the harmonics generation is of broad interest in biology.

The manuscript is clearly written and the Figures and statistical analyses are described appropriately. The graphical presentations include convincing single cell examples and, more importantly, large scale statistics (histograms and correlations for many cells). The data are directly exploited to design a heuristic model with an incoherent feedforward loop and self-inhibition. Model simulations are consistent with the intuitive interpretation of the described combinatorial regulation. The study of another reporter in Figure 6 illustrates the generality of the underlying principle.

Minor comments:

1. The final sentence of the introduction is correct but a bit mystic ("couple to multiple downstream gene circuits ...complex and unexpected gene regulation."). I am sure that the authors are aware that harmonics generation is a well-studied phenomenon in laser physics requiring simply non-linearities and high intensities. In the supplement the authors illustrate themselves that frequency doubling is even possible with Michaelis-Menten kinetics. Thus I suggest to state more clearly that multiplications of periodic signals of comparable amplitudes and appropriate phases generate harmonics (see, e.g. a supplement in Korencic PLoS One 2012). Furthermore, it might be demonstrated in the supplement more clearly how non-linearities such as Hill functions can be related to multiplications of trigonometric terms. Perhaps even the effects of high light intensities can be interpreted in that framework easily: One of the trigonometric terms dominates the other and removes frequency doubling.
2. I find "3 days of movies (10 movies in total)" confusing. The data shown in Figure 1A are 5 days long. What is the meaning of "3 days"?
3. The messages from Figure 2 are very clear but some technical aspects are unclear to me. Normally, auto-correlations are based on long-term time averages. Here only a few days are available? Thus I expect some other averages over ensembles of cells and movies. Even from the supplement I got no complete understanding. Why auto-correlations are below 1? How error bars are calculated?
4. It is a pity that even in MSB equations are omitted in the main text. I see from the supplement that a full explanation of all terms and parameters is too much but perhaps an abbreviated example of these 3 equations might be included in the main text to illustrate that the model is indeed straightforward and relatively simple.
5. Why relatively large Hill coefficients are chosen? What is the minimal Hill coefficient to get frequency doubling?

Response to Reviewer #1:

“In this manuscript Martins, Das and Locke use YFP reporters and microscopy and to examine the nature of psbAI (paralogs should be roman) expression in single Synechococcus elongatus cells “

We thank the reviewer for pointing this out and we have modified our nomenclature accordingly.

“They find that psbAI expression exhibits two peaks of gene expression at very low light that is dependent on both sigC and the circadian clock (both of which have been previously demonstrated to be required for proper expression of psbAI at the population level). These results were dependent on single cell analysis as averaging effects appear to mask this effect. Furthermore they demonstrate that rpoD6, a group 2 sigma factor, also displays a similar 2 peak expression pattern over a 24-h cycle, suggesting that 2 peaks of expression may be a more generalized mechanism. The authors also demonstrate that S. elongatus exhibits 2 peaks in cell elongation and try to make a correlation with psbA1 expression. Combined with mathematical modeling the authors put forward a model suggesting that the two peak oscillation are driven by an incoherent feedforward loop where the circadian clock activates both sigC and psbA1 and sigC in turn inhibits psbAI generating two peak oscillations in every 24-h cycle. While these data are interesting and shed light into how the circadian outputs can be modulated to generate rhythms of different periodicities there are some concerns regarding some of the data interpretation.”

We thank the reviewer for their interest in our study. We have modified the text and produced new figures to clarify our interpretation of the data, as described below.

“1. In particular there were several points of inconsistency regard the sigC results:

a. It was unclear how the traces in Figure 3 were generated for the P_{sigC}-YFP reporter. Movie 2, which showed these strains growing displayed a great deal of cell to cell variation. It was unclear as to how such variation observed in the movie resulted in the coherent and large amplitude rhythms shown in Fig. 3.”

In this and other similar figures, we plotted a mean trace across all single cells in all experiments (the dashed line) and two illustrative single cell traces from any of a number of micro-colonies (movies). We agree with the reviewer that the amplitude of the two traces we plot may not be representative of the variance in the distribution of maximal amplitudes (which is depicted in Figure EV2A, red histogram). We redrew Figure EV2 to measure the amplitudes in the same way as in Figure 1F. We chose a different example for one of the representative traces in Figure 3A, which is now shown in the figure.

The single cell time traces of Movie 2 are shown below for a little over 4 days (we did not analyse this movie any further). While the reviewer is right to point out that there is a degree of cell-to-cell variability, we do not think it paints a different picture to what we summarise in Figure 3. We have now included the traces from this and another independent movie in a new supplemental figure

(Figure S2). We hope that this new figure, in addition to the existing figure showing the distribution of amplitudes (Figure EV2A), as well as the modification to Figure 3A, will allow the reader to gain an accurate impression of the coherence of the P_{sigC} -YFP rhythms.

Data removed at authors' request

“b. The authors show that psbAI rhythms in a sigC mutant background have a single peak of expression with a 24h period oscillation. However published results (Nair et al., 2002) show period lengthening. Can the authors explain this discrepancy?”

The reviewer is correct we did not comment on the period lengthening results of *Nair et al., 2002*. These researchers found *sigC* deletions increased the period of $P_{psbA1}::luxAB$ expression by about 2 hours. To investigate whether we also observed period lengthening in the *sigC* deletion, we examined the peak to peak distances of the single frequency mode of P_{psbA1} -YFP (Figure 1B) and P_{sigC} -YFP (Figure 3B). We observe a small increase in period in the *sigC* mutants. For P_{sigC} -YFP, for example, we can measure a mean peak-to-peak distance of 24.43 hours in the WT, while in the *sigC* mutant, we measure a distance of 25.47 hours.

We now discuss this fact, and a comparison to the *Nair et al. 2002* results in the discussion (lines 310-315 in the Discussion). We suggest that in future work it will be important to examine the mode of action of possible extra components in the circuit (lines 320-322 in the Discussion). We point out that these extra components will not affect the mechanism of frequency doubling we present in this paper. The incoherent feedforward loop will still be embedded in the circuit and drive two peak oscillations.

“c. There also appeared to be some confusion about the role for SigC in their model. In particular the authors mention in lines 178-181 that the SigC auto-regulatory feedback was not necessary to observe the two peak oscillation. However, their data (Figure 1D) suggests that sigC is absolutely necessary for this observation. How are these reconciled?”

We apologise for the potentially ambiguous wording of our statement in lines 178-181 (of the previous version). The negative regulation of SigC in the system has two arms: negative feedback on itself and feedforward on *psbA1*. We mean to say that, *mathematically*, it is only the negative regulation of SigC on *itself* that is not necessary to generate two peak oscillations in *psbA1*. In our model, what is absolutely necessary to generate a two-peak oscillation is the feedforward of SigC on *psbA1*. Simulations of a *sigC* deletion are plotted in Figure 4A and are in good agreement to the experimental results of Figure 1D.

We have clarified this sentence in the text (lines 199-201 in the Results section) to make it clearer to the reader.

“2. Additionally, the authors claim that psbAI levels can account for the 2 peaks in cell elongation. However, S. elongatus displays what is known as circadian gating of cell division, where the circadian clock specifically inhibits cell division for several hours around subjective dusk. Could the observed 2-

peaks in cell elongation be due to circadian gating of cell division?” “Is this two peak in elongation rate observed in a kai mutant background?”

We thank the reviewer for noting a possible confusing element of our growth (elongation) rate measurement. We have modified the relevant section in the Materials and Methods (lines 429-430) to clarify our elongation rate measurement. We calculated the elongation rate by averaging over a sliding window at each time point. We concatenated the length time traces of cells to the traces of their mothers and daughters. In this way we can average *across* division events and obtain a continuous function for the elongation rate, which is not interrupted at the end of each cell cycle. On the other hand, as the reviewer described, gating of cell division refers to the modulation of the rate (or probability) of cell division, which fluctuates with a circadian period. While a delay (gating) in cell division will temporarily stall the birth of new cells, it does not stop cells from increasing in length. In addition to the fact that our growth rate calculations take into account division, we also observe a similar distribution of cell cycle times in the wild-type and sigC deletion mutants. Thus the gating of cell division does not appear to play a role in the two peaks of elongation that we observe.

A hallmark of gating, under certain light levels, is a bimodal distribution of cell cycle times, where one mode corresponds to the cells for which cell division was inhibited around subjective dusk (longer cell cycles) and the other mode corresponds to cells for which cell division was not inhibited (shorter cell cycles). We observe similar gating of cell division between the WT (which shows 2 peaks of elongation rate) and the sigC knock-out (which shows one), as displayed in the figure below. Hence we do not think that cell cycle gating is playing a role in our observed two peak oscillation in elongation rate.

Data removed at authors' request

“Is this two peak in elongation rate observed in a kai mutant background?”

We do not observe secondary peaks in the autocorrelation of growth in the KaiBC mutant background (see figure below). This is to be expected from our model as without the circadian clock there is no 12 or 24 hour oscillation in psbAI or other clock outputs.

Data removed at authors' request

With the reviewer's permission, we would prefer not to add these results to our manuscript, as we are preparing a different manuscript where we examine growth rates and cell division gating in greater detail, and would prefer to keep the two manuscripts separate.

“3. The authors do not mention the fact that the O'Shea lab found the sigC gene to be a direct target of RpaA, which is the transcriptional master regulator by which the Kai oscillator controls gene expression. With this point in mind, can the authors be more specific about how the two loops interact?”

We agree with the reviewer that this is an interesting and important point. We now discuss implications of those results in the text (lines 317-320 in the Discussion).

“4. Although not essential for this paper, it would have been useful to also test expression of a gene not known to be strongly affected by sigC or by environmental conditions, such as the kaiBC promoter. Data may be available that could be incorporated in the model. See:

Stability and lability of circadian period of gene expression in the cyanobacterium Synechococcus elongatus.

Clerico EM, Cassone VM, Golden SS.

Microbiology. 2009 Feb;155(Pt 2):635-41. doi: 10.1099/mic.0.022343-0.

PMID: 19202112”

We thank the reviewer for this suggestion. We added a discussion on the results of this paper and agree it will be an important factor to keep in mind going forward (lines 315-317 and 320-322 in the Discussion). In addition, please see below, in our response to major point 2 by reviewer #2, examples of times traces of PkaiBC-YFP from new experiments we carried out.

Response to Reviewer #2:

“This is an interesting paper that seeks to explain how the circadian clock in the cyanobacterium Synechococcus elongatus, which beats on a 24 h basis, is also able to generate shorter period (12 h) rhythms by frequency doubling.

The authors created a transgenic strain that expresses YFP under the control of the psbA1 promoter, a key gene involved in photosynthesis, which has been used as a circadian marker in many guises before, mostly using bioluminescence. However, previous analyses have employed population assessment of bacterial psbA1 expression, rather than single cell analyses, and therefore have not revealed 12 h oscillations previously. The paper presents compelling data, by genetic deletion, that an important component mediating frequency doubling of psbA1 is the sigC sigma factor.

Conceptually, the paper presents a new model of how frequency doubling can occur in this particular case (and is modeled well by an incoherent feedforward loop), but also potentially in other genetic circuits (e.g. mammalian clock). Therefore, the study will be of interest to researchers interested in oscillatory phenomena, including those studying circadian clocks in many organisms.”

We thank the reviewer for their interest in our study and we are glad to find the reviewer finds our work will be of general interest.

“Major points

1. The authors should discuss, and justify, why previous bioluminescence studies performed using

single cell imaging (e.g. Mihalcescu et al. Nature 2004) did not observe 12 hour rhythms, even though psbA1 drove expression of bacterial luciferase in these cells. “

We have added some lines discussing this mismatch in the main text (lines 271-276 in the Discussion). We believe that the key difference is the difference in light levels between the studies. Mihalcescu *et al.*, for example, report $100 \mu\text{E m}^{-2} \text{s}^{-1}$. We observe the two-peak phenomenon at $15 \mu\text{E m}^{-2} \text{s}^{-1}$, while at $35 \mu\text{E m}^{-2} \text{s}^{-1}$ it is largely absent. Previous studies may have therefore been conducted outside the light range where the phenomenon occurs. However, inconsistencies in measuring light at the level of the sample plane are likely to exist between different studies, so we cannot say for sure that these light levels are directly comparable. Indeed, Mihalcescu *et al.* measured an average colony doubling time of 23 h, which is of the same order of the growth rates we measured in our experiments, suggesting the “real” light levels (i.e., the flux of photons actually reaching the cells) in the two studies are closer. However, in order to measure the luminescence from the luciferase reporter Mihalcescu *et al.* had to turn the lights off for 35 minutes in between each time point sample, and so illumination was intermittently off for approximately 1/3 of the experiment (8 hours in a day). In fact, this may be the reason why their growth rates and ours are effectively similar: their cells would have grown faster when the lights are on (at high intensity), but slow down when the lights are off. By contrast, one advantage of using YFP rather than luciferase is that it means our acquisition times are much shorter (less than 5 seconds per timepoint rather than 35 minutes for luciferase). We only switch the lights off when acquiring images (turning them on when the stage travels between positions and during automatic focusing), which means lights are off for less than 3 minutes per day. The illumination conditions are therefore significantly different. Since the behaviour we observed is light-dependent, and Mihalcescu *et al.*'s cells were under a different light regime, we believe that this is the main reason why frequency doubling was not observed in their study.

“2. Related to (1), the authors should exclude a significant effect of pH in determining the phenomenon they observe. YFP fluorescence is known to be significantly affected by pH. If, for example, sigC drives a 24 h rhythm in pH (that was independent of the psbA1 rhythm), this could produce a second peak in fluorescent that was not due to psbA1 at all. One way to resolve this might be to express YFP under a constitutive promoter and see whether YFP fluorescence oscillates. An alternative would be to assay YFP protein / transcript levels (or psbA1 itself) in synchronized psbA1-YFP cells. It would be expected that if 12 h and 24 h rhythms are driven by the psbA1 promoter, then this should be seen in expression patterns, in addition to YFP fluorescence. “

If the YFP fluorescence was affected by an oscillation in pH caused by sigC we would expect to observe double peaks in fluorescence in all traces of reporters in the WT background. In the paper we already analyse P*sigC*-YFP traces in the WT background. These traces display a single peaked 24 hour period, showing evidence that the two peak phenomenon is not due to pH. However, we agree with the reviewer that this is an important point, so we carried out additional movies of circadian reporters to confirm that the two peaks are not due to an oscillation in pH. It is not straightforward to define constitutive reporters in *S. elongatus*. Owing to global circadian regulation in this organism, which ultimately drives circadian cycles of DNA supercoiling (see, for example, Smith and Williams (PNAS 2006), *Circadian rhythms in gene transcription imparted by chromosome compaction in the*

cyanobacterium Synechococcus elongatus; Woelfle et al. (PNAS 2007), *Circadian rhythms of superhelical status of DNA in cyanobacteria*; Vijayan et al. (PNAS 2012), *Oscillations in supercoiling drive circadian gene expression in cyanobacteria*) it is unlikely any promoter will have a truly flat mean fluorescence (Liu et al. (Genes Dev 1995), *Circadian orchestration of gene expression in cyanobacteria*).

Instead, we examined fluorescent reporters for the housekeeping sigma factor *rpoD1* and for the *kaiBC* operon (using a published strain from Chabot and van Oudenaarden, Nature 2007), which are not known to be regulated by *sigC*. If there was a large enough oscillation in pH in the system affecting YFP fluorescence, we would expect to see evidence of two peak oscillations in the *PkaiBC*-YFP and *PrpoD1*-YFP reporter. Below, we plot all single cell traces for three movies of representative colonies of these two reporters. The black trace represents the mean across all cells in the colony. We do not see double peaks in fluorescence in either case. For comparison, we show three representative movies of *psbAI* under the same conditions (two of which we had use in Figure EV1 as examples of synchronised and desynchronised double peaks across the whole colony). In addition to these, and as we mentioned before, we point out that we did not observe double peaks in the *sigC* reporter either (Figure 3, Figure S2).

Data removed at authors' request

“Minor points

1. Does fluorescence excitation itself affect the clock in the cyanobacterial cells? This is important since light levels influence the phenomena shown in the manuscript. Citing previous literature would be sufficient, if available. “

Fluorescence microscopy of the clock in cyanobacteria is an established technique, with reviews available describing best practice in order to avoid perturbation of the clock (e.g, Cohen SE et al., Methods Enzymology, 2015; Yokoo et al., Photosynthetic Research, 2015). We have now added references to the Materials and Methods section, with a description of how we are following best practice in the field (lines 387-390). These include avoiding fluorescent proteins in the blue regime, minimising exposure times, and spacing timepoints atleast 30 minutes apart (Yokoo et al., Photosynthetic Research, 2015).

“2. “Dampen” means “to wet”. This should be changed to “damp”.“

We thank the reviewer for this correction. Indeed, “to dampen” means “to wet”, but it also has the additional meaning of “to depress” or “to make less strong”, which was the intended meaning. However, it is true that “damp” is a more standard term in oscillatory mechanics, so we have changed our text accordingly (lines 172 in Results).

Response to reviewer #3:

“The manuscript combines in a convincing way large scale single cell imaging, appropriate reporters and mutations, network motifs and mathematical modeling. The key observations are frequency doublings (or harmonics) related to the circadian clock. In the introduction other interesting biological examples are discussed with 12 hour rhythms including circa-tidal rhythms. Thus the harmonics generation is of broad interest in biology.

The manuscript is clearly written and the Figures and statistical analyses are described appropriately. The graphical presentations include convincing single cell examples and, more importantly, large scale statistics (histograms and correlations for many cells). The data are directly exploited to design a heuristic model with an incoherent feedforward loop and self-inhibition. Model simulations are consistent with the intuitive interpretation of the described combinatorial regulation. The study of another reporter in Figure 6 illustrates the generality of the underlying principle.”

We thank the reviewer for their interest and positive words about our study.

“Minor comments:

1. The final sentence of the introduction is correct but a bit mystic (“couple to multiple downstream gene circuits ...complex and unexpected gene regulation.”). I am sure that the authors are aware that harmonics generation is a well-studied phenomenon in laser physics requiring simply non-linearities and high intensities. In the supplement the authors illustrate themselves that frequency doubling is even possible with Michaelis-Menten kinetics. Thus I suggest to state more clearly that multiplications of periodic signals of comparable amplitudes and appropriate phases generate harmonics (see, e.g. a supplement in Korencic PLoS One 2012). “

We thank the reviewer for pointing out the generality of frequency doubling in other fields. We added a sentence on the multiplication of sinusoidal factors to the introduction where we now reference Korencic *et al.*, *PLoS One* 2012 (lines 49-51). We have also deleted the adjective unexpected from the last sentence of the introduction (line 91).

“Furthermore, it might be demonstrated in the supplement more clearly how non-linearities such as Hill functions can be related to multiplications of trigonometric terms. Perhaps even the effects of high light intensities can be interpreted in that framework easily: One of the trigonometric terms dominates the other and removes frequency doubling.”

We agree with the reviewer that it is a good idea to show the reciprocity between our model functions and previous work on second harmonics arising from multiplications of trigonometric terms. We added a new sub-section (Section III.2) to the Appendix supplement. In it, we derive and present a general form for the decomposition of two-input Hill functions (where the inputs are sinusoids) into a combination of first and second harmonic terms.

“2. I find “3 days of movies (10 movies in total)” confusing. The data shown in Figure 1A are 5 days long. What is the meaning of “3 days”?”

We agree with the reviewer that our wording is unclear. By “3 days of movies” we meant to say 3 independent experiments (started at 3 different days from 3 different cell cultures). From these we analysed 10 movies (10 micro-colonies). We modified the way this information is presented in the text (line 103 in Results).

“3. The messages from Figure 2 are very clear but some technical aspects are unclear to me. Normally, auto-correlations are based on long-term time averages. Here only a few days are available? Thus I expect some other averages over ensembles of cells and movies. Even from the supplement I got no complete understanding.”

We agree with the reviewer that the technical details on the cross-correlations can be made clearer. As mentioned in the Materials and Methods (line 434), we followed the method introduced by Dunlop et al. (Nat Gen, 2008) to compute cross-correlations on time series in tree structures. Equation (7) computes the cross-correlation function in a *micro-colony* (movie), in which cells branch out of each other – in a lineage tree – from a single founder.

In a first step, all N_c lineages in the tree are extracted, typically one for each cell present at the end of the movie. One therefore obtains N_c time series and calculates the cross-correlation between a pair of variables in each of these lineages in the usual way, i.e., for each time lag, one computes $\sum_{\text{all data points}} x(n + \tau)y(n)$. Secondly, all cross-correlations thus obtained are averaged across all cells/lineages. However, because in a tree structure many pairs of data points are common to more than one lineage, the contributions of repeated pairs of data points are subtracted to ensure each pair is only counted once, hence the negative term in equation (7). Finally, for each strain, we average the cross-correlations across all movies (where one movie tracks one tree), and that final average is what is plotted in the panels of Figure 2.

We have rewritten the relevant section in the Materials and Methods (lines 434-455) to include more details and a clearer rationale.

“Why auto-correlations are below 1? How error bars are calculated?”

The reviewer is right to be confused about the maximal value of the auto-correlation not averaging to 1. It should be 1. We originally made a mistake where, when averaging across all movies, divided by the incorrect number of movies in the denominator. We apologise for this mistake, but fortunately it only introduces a scaling error, which does not affect the shape of the cross-correlation profiles and their underlying message. We thank the reviewer for this observation, and we have corrected the figure accordingly.

The error bars represent standard errors of the mean, where the number of movies is the number of samples. This is mentioned in the figure legend (lines 690-691).

“4. It is a pity that even in MSB equations are omitted in the main text. I see from the supplement that a full explanation of all terms and parameters is too much but perhaps an abbreviated example of these 3 equations might be included in the main text to illustrate that the model is indeed straightforward and relatively simple. “

We agree with the reviewer’s suggestion and have now added a simplified version of the model equations to the main text (lines 185-194 in Results).

“5. Why relatively large Hill coefficients are chosen? What is the minimal Hill coefficient to get frequency doubling? “

The simulations shown in Figure 4 are meant to be an illustration of the dynamical output of the model in comparison to the observed expression data, not a best fit of the data. As such, the parameters are somewhat arbitrary. The reviewer is, of course, correct to draw attention to the Hill coefficients because these generate further multiplications of periodic terms. As the reviewer pointed out above, such terms are critical in generating and modulating the amplitude of double frequency harmonics. As such, higher Hill coefficients can generally be expected to either generate double peaks of higher prominence or expand the parameter space where they are found. In order to explore how frequency doubling is affected by the Hill coefficient choice, we used equation (S14), $f = V \frac{(u_1)^{h_1}}{1+(u_1)^{h_1}+(u_2)^{h_2}+(u_1)^{h_1} (u_2)^{h_2}}$, to calculate the production rate of a target regulated by two sinusoidal inputs. We sampled 100,000 sets of parameters from uniform distributions and determined whether the production rate was two peaked or single peaked. Finally, if two peaked, we calculated the relative prominences (vertical distance between the top of the peak and its base) of the two peaks. The product of the two prominences is therefore a proxy for the “strength” of the frequency doubling. The distribution of Hill coefficients in sets where the product of relative prominences is greater than 0.04 (in Figure 4B, the equivalent product is 0.042) is shown below (panel A). Clearly, higher Hill coefficients are favoured, as one would intuitively expect, but combinations with lower Hill coefficients are not prohibited. Below, in panel B, we present a comparison between the simulation with the full model shown in Figure 4B and a simulation for the same set of parameters, but where the Hill coefficients of 5 were reduced to 3. We can include these panels in the Appendix supplement, if the reviewer feels they will enhance it.

In the Appendix, we show that frequency doubling is detectable in the production rate of a deterministic system even if the Hill coefficients are chosen to be 1 (because in our model, there is competition between the two regulators, which yields a term that expresses the product of the two). However, in order for frequency doubling in the production rate to be detectable at the level of expression when the Hill coefficients are low, fast dilution-degradation rates are required.

Data removed at authors’ request

“3. The messages from Figure 2 are very clear but some technical aspects are unclear to me. Normally, auto-correlations are based on long-term time averages. Here only a few days are available? Thus I expect some other averages over ensembles of cells and movies. Even from the supplement I got no complete understanding.”

We agree with the reviewer that the technical details on the cross-correlations can be made clearer. As mentioned in the Materials and Methods (line 434), we followed the method introduced by Dunlop et al. (Nat Gen, 2008) to compute cross-correlations on time series in tree structures. Equation (7) computes the cross-correlation function in a *micro-colony* (movie), in which cells branch out of each other – in a lineage tree – from a single founder.

In a first step, all N_c lineages in the tree are extracted, typically one for each cell present at the end of the movie. One therefore obtains N_c time series and calculates the cross-correlation between a pair of variables in each of these lineages in the usual way, i.e., for each time lag, one computes $\sum_{\text{all data points}} x(n + \tau)y(n)$. Secondly, all cross-correlations thus obtained are averaged across all cells/lineages. However, because in a tree structure many pairs of data points are common to more than one lineage, the contributions of repeated pairs of data points are subtracted to ensure each pair is only counted once, hence the negative term in equation (7). Finally, for each strain, we average the cross-correlations across all movies (where one movie tracks one tree), and that final average is what is plotted in the panels of Figure 2.

We have rewritten the relevant section in the Materials and Methods (lines 434-455) to include more details and a clearer rationale.

“Why auto-correlations are below 1? How error bars are calculated? “

The reviewer is right to be confused about the maximal value of the auto-correlation not averaging to 1. It should be 1. We originally made a mistake where, when averaging across all movies, divided by the incorrect number of movies in the denominator. We apologise for this mistake, but fortunately it only introduces a scaling error, which does not affect the shape of the cross-correlation profiles and their underlying message. We thank the reviewer for this observation, and we have corrected the figure accordingly.

The error bars represent standard errors of the mean, where the number of movies is the number of samples. This is mentioned in the figure legend (lines 690-691).

“4. It is a pity that even in MSB equations are omitted in the main text. I see from the supplement that a full explanation of all terms and parameters is too much but perhaps an abbreviated example of these 3 equations might be included in the main text to illustrate that the model is indeed straightforward and relatively simple. “

We agree with the reviewer’s suggestion and have now added a simplified version of the model equations to the main text (lines 185-194 in Results).

“5. Why relatively large Hill coefficients are chosen? What is the minimal Hill coefficient to get frequency doubling? “

The simulations shown in Figure 4 are meant to be an illustration of the dynamical output of the model in comparison to the observed expression data, not a best fit of the data. As such, the parameters are somewhat arbitrary. The reviewer is, of course, correct to draw attention to the Hill coefficients because these generate further multiplications of periodic terms. As the reviewer pointed out above, such terms are critical in generating and modulating the amplitude of double frequency harmonics. As such, higher Hill coefficients can generally be expected to either generate double peaks of higher prominence or expand the parameter space where they are found. In order to explore how frequency doubling is affected by the Hill coefficient choice, we used equation (S14), $f = V \frac{(u_1)^{h_1}}{1+(u_1)^{h_1}+(u_2)^{h_2}+(u_1)^{h_1}(u_2)^{h_2}}$, to calculate the production rate of a target regulated by two sinusoidal inputs. We sampled 100,000 sets of parameters from uniform distributions and determined whether the production rate was two peaked or single peaked. Finally, if two peaked, we calculated the relative prominences (vertical distance between the top of the peak and its base) of the two peaks. The product of the two prominences is therefore a proxy for the “strength” of the frequency doubling. The distribution of Hill coefficients in sets where the product of relative prominences is greater than 0.04 (in Figure 4B, the equivalent product is 0.042) is shown below (panel A). Clearly, higher Hill coefficients are favoured, as one would intuitively expect, but combinations with lower Hill coefficients are not prohibited. Below, in panel B, we present a comparison between the simulation with the full model shown in Figure 4B and a simulation for the same set of parameters, but where the Hill coefficients of 5 were reduced to 3. We can include these panels in the Appendix supplement, if the reviewer feels they will enhance it.

In the Appendix, we show that frequency doubling is detectable in the production rate of a deterministic system even if the Hill coefficients are chosen to be 1 (because in our model, there is competition between the two regulators, which yields a term that expresses the product of the two). However, in order for frequency doubling in the production rate to be detectable at the level of expression when the Hill coefficients are low, fast dilution-degradation rates are required.
Data.removed.at.authors'.request.

Thank you for submitting your revised study. We have now heard back from the three referees who were asked to evaluate your manuscript. As you will see below, the referees think that all issues have been satisfactorily addressed and they support publication of the study. Before we formally accept the study for publication, we would ask you to address some minor editorial issues listed below.

We have recently implemented a "model curation service" for papers that contain mathematical models. This is done together with Prof. Jacky Snoep and the FAIRDOM team. In brief, the aim is to enhance reproducibility and add value to papers containing mathematical models. As you will see in the Model Curation Report (pasted below the referee reports) there are some minor issues, which we would ask you to fix when you submit your revised manuscript.

REFeree REPORTS

Reviewer #1:

The authors have addressed our concerns in the revised manuscript.

Reviewer #2:

This revised manuscript addresses all of the points raised previously both with experimental data and clarifications in the text.

Reviewer #3:

The authors revised the manuscript carefully and answered all my mainly technical concerns appropriately.

MODEL CURATION REPORT

The model file for MSB-16-7087R was provided in SBML format, and described the model in rate-laws. The model file is correct SBML and could be directly simulated. For inclusion in the JWS Online database, we converted the model from the rate-law description into a process description (i.e. the ODEs are split up in production and consumption processes). This was done to be able to make a schema for the model structure lay-out in the simulator.

I tested the reproduction of the simulation shown in the manuscript: i.e. Fig. 4B, Fig. 6C and Fig 6D. For each of the figures the authors provided the specific simulation settings in the supplementary materials. Most of the simulations could be reproduced, but there are still a number of small issues that need to be addressed.

The authors did not list the initial conditions for any of the simulations. I ran the simulations with initial conditions for all variables equal to 0. This leads to differences in published simulation results compared to the results shown on JWS Online. I see that the authors chose the initial conditions such that they fall on the limit cycle. If the authors supply the initial conditions in the supplementary material, then the models simulations can be made to more precisely reproduce the simulations in the manuscript.

Fig 4b and 6c could be reproduced.

For the output insert shown in Fig 6d, we could reproduce the panels b,c,d but not panel a. I suspect that the value for m (listed in supplementary material as 0.08) is wrong. When I simulate the model with $m=0.8$ the simulation result is much closer to what is shown in Fig. 6D.

Please check, and correct the parameter values for the reproduction of the first output signal shown in Fig 6d (panel a in supplementary material; lines 182-183).

Fig EV3 A could be reproduced, although it appears that there is a shift in y-axis. Please check your simulations against the results in JWS Online (see below). For the simulations I only change the parameter kb from 0.1 to 10 (all other parameter values have default values).

Fig EV3 B could not be reproduced. For the simulation I only change the parameter Tc to 9.3 (all other parameter values have default values). Please specify how Figure EV3B was produced, i.e. specify what model variables correspond to the SigC low light and SigC high light lines. Please specify the parameters that were changed compared to the default model (if anything more than parameter Ty).

2nd Revision - authors' response

17 November 2016

Thank you for your positive assessment of our manuscript. Below we address the final editorial and modelling issues that you have brought to our attention.

“The authors did not list the initial conditions for any of the simulations. I ran the simulations with initial conditions for all variables equal to 0. This leads to differences in published simulation results compared to the results shown on JWS Online. I see that the authors chose the initial conditions such that they fall on the limit cycle. If the authors supply the initial conditions in the supplementary material, then the models simulations can be made to more precisely reproduce the simulations in the manuscript.”

Prof. Snoep is correct in that we should have specified initial conditions. Our initial conditions are 0 for all species in all figures, but we extracted and show only the final 120 h of a much longer simulation (typically, 600 h although steady state is reached much earlier, after just one or two cycles). This information is now explicitly stated in the model section in the Appendix.

“For the output insert shown in Fig 6d, we could reproduce the panels b,c,d but not panel a. I suspect that the value for m (listed in supplementary material as 0.08) is wrong. When I simulate the model with m=0.8 the simulation result is much closer to what is shown in Fig. 6D. Please check, and correct the parameter values for the reproduction of the first output signal shown in Fig 6d (panel a in supplementary material; lines 182-183).”

Prof. Snoep is right to point out the information we provided does not allow reproduction of the figure, and that this is due to a typo in the value of parameter m. The value is actually correct, but the parameter is named incorrectly – the parameter we modified was b, not m. It should therefore read b=0.08. There was also a shift of 0.25 units in the vertical axis (Figure S7A), which we now fixed. We apologise for these errors.

“Fig EV3 A could be reproduced, although it appears that there is a shift in y-axis. Please check your simulations against the results in JWS Online (see below). For the simulations I only change the parameter kb from 0.1 to 10 (all other parameter values have default values).”

In Figure EV3A (high light condition) two parameters were changed, kb (from 0.1 h⁻¹ to 10 h⁻¹) and Tc (19.5 h to 9.3 h). This is stated in lines 174-176 of the Appendix (2nd paragraph after Table S1). However, in the SBML file the name Tc was used for a different parameter: the period of the clock, Tc = 24 h. The parameter referred to as Tc in the text actually referred to Td in the code file (default value of 19.5 h). For consistency, we now renamed this parameter, Td, in the text and in Table S1. Changing kb alone does indeed result in the trace displayed in JWS Online, but changing both parameters reproduces our figure.

“Fig EV3 B could not be reproduced. For the simulation I only change the parameter Tc to 9.3 (all other parameter values have default values). Please specify how Figure EV3B was produced, i.e. specify what model variables correspond to the SigC low light and SigC high light lines. Please

specify the parameters that were changed compared to the default model (if anything more than parameter Ty).“

We thank Prof. Snoep for picking up a potential source of confusion in this panel. We had not stated what form of SigC is being plotted in this panel. We are plotting the sum of the active and inactive forms of SigC (in the SBML file nomenclature, SigC + SigC_a (1)), because this is the only quantity we can directly relate to the transcriptional reporter we use in the experiments. This is now stated in the figure legend and also in the Appendix text.

(1) For consistency, we renamed the active form – from to – in the text.

As for the parameters, in the high light case (dashed line) these are the same as in Figure EV3A (i.e., both kb modified to 10 h⁻¹ and Td (see above for change in nomenclature) modified to 9.3 h. All other parameters are the default parameters). In the low light case (solid line), the parameters are set as in Figure 4.

We have modified the text to include specific references to individual panels and traces, in order to make these parameter changes clearer to the reader.

Finally, in the SBML file, we renamed all parameters whose names were not consistent with how they were named in the text.

3rd Editorial Decision

23 November 2016

Thank you again for sending us your revised manuscript. I have circulated to Prof. Snoep the corrections related to the model and he has confirmed that all figures can be reproduced with this additional information. As such, I am pleased to inform you that your paper has been accepted for publication.

Corresponding Author Name: James C. W. Locke

Manuscript Number: MSB-16-7087